# Clinical and Molecular Features in Medulloblastomas Subtypes in Children in a Cohort in Taiwan

**DOI:** 10.3390/cancers14215419

**Published:** 2022-11-03

**Authors:** Kuo-Sheng Wu, Shian-Ying Sung, Man-Hsu Huang, Yu-Ling Lin, Che-Chang Chang, Chia-Lang Fang, Tai-Tong Wong, Hsin-Hung Chen, Min-Lan Tsai

**Affiliations:** 1Graduate Institute of Clinical Medicine, College of Medicine, Taipei Medical University, Taipei 110, Taiwan; 2International Ph.D. Program for Translational Science, Taipei Medical University, Taipei 110, Taiwan; 3The Ph.D. Program for Translational Medicine, Taipei Medical University, Taipei 110, Taiwan; 4Department of Pathology, Shuang-Ho Hospital, Taipei Medical University, New Taipei City 235, Taiwan; 5Agricultural Biotechnology Research Center, Academia Sinica, Taipei 115, Taiwan; 6Department of Pathology, School of Medicine, College of Medicine, Taipei Medical University, Taipei 110, Taiwan; 7Department of Pathology, Taipei Medical University Hospital, Taipei Medical University, Taipei 110, Taiwan; 8Division of Pediatric Neurosurgery, Department of Neurosurgery, Taipei Medical University Hospital, Taipei Medical University, Taipei 110, Taiwan; 9Pediatric Brain Tumor Program, Taipei Cancer Center, Taipei Medical University, Taipei 110, Taiwan; 10Neuroscience Research Center, Taipei Medical University Hospital, Taipei 110, Taiwan; 11Division of Pediatric Neurosurgery, The Neurological Institute, Taipei Veterans General Hospital and School of Medicine, National Yang Ming Chiao Tung University, Taipei 112, Taiwan; 12Department of Pediatrics, School of Medicine, College of Medicine, Taipei Medical University, Taipei 110, Taiwan; 13Department of Pediatrics, College of Medicine, Taipei Medical University Hospital, Taipei Medical University, Taipei 110, Taiwan

**Keywords:** medulloblastoma, molecular subgroups, subtypes, RNA sequencing, DNA methylation array, M2 macrophages

## Abstract

**Simple Summary:**

Medulloblastoma (MB) was classified into four subgroups: WNT, SHH, group 3, and group 4. In 2017, 12 subtypes within 4 subgroups and 8 subtypes within non-WNT/non-SHH subgroups according to the heterogenous features were announced. In this study, we aimed to identify the heterogeneity of molecular features for discovering subtype specific factors linked to diagnosis and prognosis. We retrieved 70 MBs to perform RNA sequencing and a DNA methylation array. Integrated with clinical annotations, we classified 12 subtypes of pediatric MBs. We found that M2 macrophages were enriched in SHH β, which correlated with good outcomes of SHH MBs. The high infiltration of M2 macrophages may be an indicator of a favorable prognosis and therapeutic target for SHH MBs. Furthermore, C11orf95-RELA fusion was observed to be associated with recurrence and a poor prognosis. These results will contribute to the establishment of a molecular diagnosis linked to prognostic factors of relevance for MBs.

**Abstract:**

Medulloblastoma (MB) was classified into four molecular subgroups: WNT, SHH, group 3, and group 4. In 2017, 12 subtypes within 4 subgroups and 8 subtypes within non-WNT/non-SHH subgroups according to the differences of clinical features and biology were announced. In this study, we aimed to identify the heterogeneity of molecular features for discovering subtype specific factors linked to diagnosis and prognosis. We retrieved 70 MBs in children to perform RNA sequencing and a DNA methylation array in Taiwan. Integrated with clinical annotations, we achieved classification of 12 subtypes of pediatric MBs in our cohort series with reference to the other reported series. We analyzed the correlation of cell type enrichment in SHH MBs and found that M2 macrophages were enriched in SHH β, which related to good outcomes of SHH MBs. The high infiltration of M2 macrophages may be an indicator of a favorable prognosis and therapeutic target for SHH MBs. Furthermore, C11orf95-RELA fusion was observed to be associated with recurrence and a poor prognosis. These results will contribute to the establishment of a molecular diagnosis linked to prognostic indicators of relevance and help to promote molecular-based risk stratified treatment for MBs in children.

## 1. Introduction

Medulloblastoma (MB) is a common malignant brain tumor in children. Demographics, clinical information, and molecular data are significantly predictive factors for survival. According to the 2016 WHO classification, four molecular subgroups: WNT, SHH, group 3 (G3), and group 4 (G4) are included in MBs [1]. The subtypes within the molecular subgroups are defined as 12 subtypes [2]. G3 and G4 are merged as non-WNT/non-SHH MBs and comprised of eight subtypes by Northcott et al. [3]. The diversity of clinical features, demographics, and genetic and cytogenetic aberrations exists in MB subtypes. Two subtypes are included in the WNT subgroup: α and β, which exhibit favorable outcomes. WNT α mainly exists in children and presents with monosomy 6. Four subtypes are included in the SHH subgroup: α, β, γ, and δ, with different age distributions. SHH α presents in children and has the following features: TP53 mutations; focal amplifications in MYCN, GLI2, and YAP1; and broad loss in 9q, 10q, 17p. SHH β presents in infants and is associated with a high metastatic rate. SHH β presents the worst outcomes, which is associated with focal PTEN deletion. SHH γ presents in infants and is enriched histologically by MBEN, which indicates favorable outcomes. SHH δ mainly presents in adults and shows a favorable outcome as SHH γ.

Recently, two independent studies have announced various subtypes in non-WNT/non-SHH MBs. Cavalli and colleagues identified the G3 (α, β, γ) and G4 (α, β, γ) subtypes [2]. Northcott and colleagues identified eight subtypes (I to VIII) in non-WNT/non-SHH MBs, which were recruited in the 2021 WHO CNS5 classification [3,4]. Usually, subtype II to IV belong to G3 and subtype V to VIII belong to G4 [5]. Subtype I represents the least common subtype, whereas subtype VIII is the most common and only consists in G4 [3,6]. Generally, no chromosome aberrations are found in subtype I, while i17q are enriched in subtype VIII [6]. MYC amplification is enriched in subtype II and III and is associated with poor outcomes (5-year survival: 50% in subtype II, 43% in subtype III) [5]. Subtype VII is associated with a favorable 5-year survival (85%) [6].

Gene expression and DNA methylation profiles are the current standard for MB subgrouping and subtyping. Recently, the similarity network fusion (SNF) method for clustering was proposed [7]. By integrating gene expression and DNA methylation data, MB subgroups can divide into various subtypes [2]. In the previous study, we collected childhood MBs to identify a molecular–clinical correlation and defined an adjusted Heidelberg risk stratification scheme for treatment protocol guidelines in multiple centers in Taiwan [8]. Different MB subtypes need to be classified based on molecular and clinical heterogeneity for establishing molecular diagnostic and prognostic markers.

In this study, we retrieved 70 childhood MBs to perform RNA sequencing (RNA-Seq) and a DNA methylation array to perform subtype clustering in Taiwan. Integrated with clinical annotations, we achieved classification of 12 subtypes of pediatric MBs in our cohort series with reference to the other reported series. We characterized high infiltration of M2 macrophages in SHH β, which may be an indicator of a favorable prognosis and a therapeutic target for SHH MBs. Furthermore, C11orf95-RELA fusion was observed and associated with recurrence and poor outcomes. These results will contribute to the establishment of a molecular diagnosis linked to prognostic factors of relevance and further help to promote molecular-based, risk-stratified treatment for MBs in children.

## 2. Materials and Methods

### 2.1. Patient Cohort 

There were 70 MB cases collected from Taipei Veterans General Hospital (Taipei VGH) and Taipei Medical University Hospital (TMUH), retrieved between 1989–2019, in children. Among MB cases, there were 64 primary tumors, 5 first recurrence, and 1 metastasis. All subjects gave written informed consent in accordance with the Declaration of Helsinki. The samples were fully encoded and used under a protocol approved by the Institutional Review Board of Human Subjects Research Ethics Committee of the Taipei Medical University Hospital and Chang Gung Memorial Hospital, Taiwan (IRB approval number 201701441A3).

### 2.2. Retrieve of Clinical Data

The retrieved clinical data included age, sex, metastasis status, histological variant, follow-up, and death. The centers of the tumor locations were defined as midline of the fourth ventricle (Midline/4thV) and cortex-centered, including cerebellar vermis (CV), cerebellar hemisphere (CH), and cerebellar pontine angle (CPA) location tumors. We defined the status of metastasis at diagnosis as M0-1 and M2-3 according to Chang’s operative staging system [9].

### 2.3. Gene Expression Profiles by RNA-Seq

RNA-Seq was performed as described in the previous study [8]. Briefly, RNA-Seq was run in a Nextseq 500 sequencing instrument (Illumina) for paired-end reads. Gene expression tables were extracted by Kallisto [10] and the tximport [11] package in the R environment. The RNA-Seq data of 70 MB cases are available in the Gene Expression Omnibus (GSE143940 and GSE158413).

### 2.4. Applying RNA-Seq to Identify Mutations

There were 73 clinically relevant mutations selected for mutations that were detected in this cohort series. These selected mutations were linked to DNA damage response (DDR), MB genesis, a genetic predisposition for MB, the MAPK and PI3K/mTOR pathways, and pediatric cancer predisposition syndromes [3,8,12,13,14,15,16]. RNA-Seq raw data were aligned using HISAT2 [17], followed by variant calling using the HaplotypeCaller tool in GATK. Variants were annotated using ANNOVAR [18] based on COSMIC database [19], and all variants in IGV with alignment level were visualized [20].

### 2.5. Immune Cell Deconvolution

Cell type deconvolutions were estimated as described in the previous study [21]. Briefly, the scores of 64 cell types in 5 major cell populations were computed with the gene expression data set normalized to TPM level of 489 cell population specific markers with xCell [22]. The scores of 34 immune cell types were compared between MB subtypes. The resulting scores are presented in arbitrary units.

### 2.6. DNA Methylation Array Profiling

The DNA methylation array was performed as described in the previous study [21]. Raw data files were read and preprocessed using the capabilities of Minfi [23] and the ChAMP [24] package in the R environment.

### 2.7. Applying DNA Methylation Profiles to Identify Copy Number Variations

The genetic status of chromosomes or selected genes was deciphered from the methylation array data. Selected copy number variations were identified from array data by using the conumee package in the R environment, as previously described [2,25,26]. The log2 ratio of chromosomes or genes more than 0.2 was defined as gain (amplification), and that of less than −0.2 was defined as loss (deletion).

### 2.8. Similarity Network Fusion (SNF) Analysis for WNT and SHH Subtype Clustering

The SNF method was performed in the cohort series as described in the previous study [21]. Briefly, subtype clustering was performed by the SNFtool package in the R environment based on the top 1% of the most differentially expressed common genes (*n* = 216) and probes (*n* = 3211) from a previous study [2]. The parameters of SNF were referred to the previous study [21].

### 2.9. Random Forest (RF) for Non-WNT/Non-SHH Subtype Clustering

The subtyping of non-WNT/non-SHH MBs was based on a web-based classifier of MB G3/4 subgroups (https://www.molecularneuropathology.org/mnp, accessed on 14 August 2021), as described in the previous study [21]. Briefly, Illumina Infinium MethylationEPIC array raw signal IDAT-files were uploaded and normalized by a two-factor linear model on log2 transition to the web-based classifier with adjustment for frozen derivatives and patient gender. The most differential 50,000 CpG loci were implemented to calculate a RF score between 0 and 1 with multinomial logistic regression for non-WNT/non-SHH subtypes prediction [27].

### 2.10. Survival Analysis

Overall survival (OS) analysis was based on the date of first tumor surgery (diagnosis date), last follow-up, and death. OS analysis based on the scores of various cell types or the expression of genes was performed by the Kaplan–Meier method by using the surv_cutpoint function within the survminer package in the R environment. The differences of survivals were assessed using the log-rank test. The association between categorized variables was determined by the Kruskal–Wallis test. A *p* value < 0.05 was considered statistical significance.

## 3. Results

### 3.1. Subsection of Molecular Subgroups Were Identified by Integrative Gene Expression and DNA Methylation Profiles

We retrieved 70 pediatric MBs to perform RNA-Seq and 66 of this cohort for the DNA methylation array. By clustering analysis, three established molecular subgroups were identified: WNT (*n* = 8, 11.4%), SHH (*n* = 24, 34.3%), and non-WNT/non-SHH (*n* = 38, 54.3%) (Figure 1a,b). There were 20 cases (28.6%) with metastasis at diagnosis in this cohort. SHH presented the highest recurrent rate (*n* = 11, 45.8%), which correlated with the worst prognosis (5-year overall survival (OS) after recurrence: 12.8%). The male/female ratio was 1 in all MBs; however, the ratio was 0.1 in WNT (Figure 1c). The distributions of age at diagnosis were different in three subgroups (Figure 1d). Most of the WNT and SHH cases displayed classic pathology (WNT: *n* = 5, 62.5%, SHH: *n* = 12, 50%), while half of the non-WNT/non-SHH cases displayed LCA pathology (*n* = 19) (Figure 1e). Most MBs were located in the midline of the fourth ventricle, including all WNT (Figure 1f). The high frequency of metastasis existed in non-WNT/non-SHH (*n* = 16, 42.1%) (Figure 1g). SHH MBs demonstrated the worst outcome (Figure 1h,i).

We combined gene expression and the DNA methylation profile to perform subtype clustering. In this cohort series, MBs were classified into WNT (α, β), SHH (α, β, γ), and non-WNT/non-SHH (II to VIII) (Appendix A). The characteristics including gender, age, histological variants, tumor location, metastasis status, survival, cytogenetic, genetic aberrations, and immune cell enrichment of subtypes were identified (Appendix A). We further compared demographics and clinical annotations of SHH and non-WNT/non-SHH subtypes in our and SickKids cohorts (Appendix A).

### 3.2. Characteristics in Subtypes of WNT MBs

By integrative clustering analysis, WNT α (*n* = 7, 87.5%) and WNT β (*n* = 1, 12.5%) were clustered in this cohort (Figure 2a,b). The only one male in WNT was classified as α subtype (Figure 2c). The median age at diagnosis was 8.4 years (range, 4 to 11.4 years) in WNT α (Figure 2d). Most of WNT α (*n* = 4, 57.1%) displayed classic pathology (Figure 2e). All WNT presented no metastasis, which related to a very good survival rate at 100% (Figure 2f). In our cohort series, monosomy 6 was found in all WNT α but not in WNT β (Figure 2g). The CTNNB1 mutation was observed in all WNT, and the DDX3X mutation was observed in 42.9% (*n* = 3) of WNT α (Figure 2h). The TP53 mutation (c.G818A; p.R273H) was observed in one WNT α patient. PVT1 fusion was found in WNT (*n* = 2, 25%) (Appendix A).

### 3.3. Characteristics in Subtypes of SHH MBs

We identified three subtypes: SHH α (*n* = 7, 29.2%), SHH β (*n* = 9, 37.5%), and SHH γ (*n* = 8, 33.3%) in this cohort due to the enrollment of pediatric patients only (Figure 3a,b). SHH δ mainly presents in adults among SHH subtypes [2]. The proportion of male and female is nearly equal in SHH MBs (Figure 3c). SHH α and β typically occurs in children with a median age at 6.5 and 4.1 years, whereas SHH γ mainly occurs in infants (age < 2 years old: *n −* 5, 62.5%) with a median age at 1.4 years (Figure 3d). Classic pathology mainly presents in SHH β (*n* = 6, 75%) (Figure 3e). Notably, MBEN histology only presented in SHH γ in our and other cohorts [2]. The localization of tumors was diverse among SHH subtypes. Half of the SHH α tumors were located in the cerebellar hemisphere (CH) (*n* = 3), while SHH β was mainly located in the midline of the fourth ventricle (*n* = 8, 88.9%) (Figure 3f). Tumor metastasis occurred in both SHH α (*n* − 1, 16.7%) and SHH β (*n* − 3, 33.3%) but not in SHH γ (Figure 3g). The highest metastatic rate presented in SHH β in our and other studies [2,6]. However, SHH γ showed the worst outcomes (5-year survival: 62.5%) (Figure 3h) and survival after tumor recurrence (1-year survival: 25%) (Figure 3i). A previous study reported that the loss of chromosome 14q was a risk factor for SHH MBs [28]. The loss of 14q was present in SHH α (*n* = 1, 14.3%) and β (*n* = 2, 22.2%), which was associated with poor outcomes (Figure 3j). We found broad chromosome loss in 9q (*n* = 2, 28.6%) and 10q (*n* = 3, 42.9%) in SHH α, which was consistent with a previous study [2]. In addition, we also found the loss of 9q (*n* = 2, 25%) in SHH γ. In terms of gene coverage, PTEN deletion was found in SHH α (*n* = 3, 42.9%) and γ (*n* = 1, 12.5%) (Figure 3k). MYCN or GLI2 amplification was observed in SHH α (*n* = 4, 50%), and the co-amplification was observed in 3 of 4 cases (Figure 3k). The focal amplifications of MYCN and GLI2 commonly occurred in SHH α [2]. Somatic TP53 mutation (c.G818A; R273H) existed in one SHH α and was correlated with poor outcomes [8]. The PTCH1 mutation mainly existed in SHH γ (*n* = 4, 50%), and the NOTCH2 mutation only existed in SHH β (*n* = 3, 33.3%) (Figure 3l). The mutation involved in the SHH pathway was enriched in SHH γ (*n* = 5, 62.5%) (Figure 3m). C11orf95-RELA fusion was found in one SHH β (Appendix A). Cell type enrichment analysis was performed as in the previous study [21]. M2 macrophages were enriched in SHH β (Figure 3n) and were correlated with good outcomes of SHH MBs (Figure 3o). To validate whether M2 macrophage enrichment is specific to SHH β, we also analyzed public data with 115 pediatric SHH MBs from the SickKids cohort study. M2 macrophages were enriched in SHH β (Appendix A), which correlated with good outcomes of SHH MBs in the SickKids cohort (Appendix A). A high expression of CCL2 was observed in SHH β in our and the SickKids cohort (Appendix A and Appendix A). CCL2 is frequently overexpressed in tumor cells in the tumor microenvironment (TME) for recruiting tumor-associated macrophages (TAMs) to support tumor growth [29]. Highly expressed M2 macrophage relevant genes: CD68, CD163, CD204 (MSR1), CD206 (MRC1), CD209, CSF1R, and Dectin-1 (CLEC7A) were validated in SHH β in our and the SickKids cohort (Appendix A and Appendix A). Furthermore, high expressions of above genes were correlated with favorable outcomes in our and the SickKids cohort (Appendix A and Appendix A).

### 3.4. Characteristics in Subtypes of Non-WNT/Non-SHH MBs

There were 7 distinct subtypes in the non-WNT/non-SHH subgroup: II (*n* = 4, 10.5%), III (*n* = 3, 7.9%), IV (*n* = 4, 10.5%), V (*n* = 1, 2.6%), VI (*n* = 5, 13.2%), VII (*n* = 13, 34.2%), and VIII (*n* = 5, 13.2%) in this cohort (Figure 4a). Females mainly exhibited subtype II (*n* = 3, 75%) and VIII (*n* = 3, 60%), whereas no females exhibited subtype III (Figure 4b). The highest median age at diagnosis presented in subtype VIII (10.1 years), while the lowest presented in subtype IV (3 years) (Figure 4c). LCA pathology mainly existed in subtype II and VII, and classic pathology mainly existed in subtype IV, VI, and VIII (Figure 4d). Moreover, there was one MB with melanotic myogenic differentiation (MM) found in subtype III. Most non-WNT/non-SHH tumors were located in the midline of the fourth ventricle (Figure 4e). Interestingly, one subtype VII was located at the cerebellar pontine angle (CPA). The highest metastatic rate presented in subtype II and IV (50%), whereas the lowest presented in subtype VIII (20%) (Figure 4f). The worst outcomes and overall survival after recurrence presented in subtype II (5-year survival: 0%), whereas favorable outcomes presented in subtype III, IV, and VIII (Figure 4g,h). The board cytogenetic aberrations are summarized in Figure 4i. Chromosome 7 gain was found in subtype VI (*n* = 5, 80%), and chromosome 8 loss was found in subtype VI (*n* = 3, 60%) and subtype VII (*n* = 8, 61.5%). Chromosome 17p loss was found in all subtype VIII (*n* = 5) and 75% of subtype II (*n* = 3). Isochromosome 17q (i17q) was found in subtype VIII (*n* = 2, 40%) and subtype VI (*n* = 1, 20%). The focal MYC amplification was enriched in subtype II, as reported by Northcott et al. [3] (Figure 4j and Appendix A). MYCN and CDK6 amplification were enriched in subtype VI (Figure 4j). Mutations in a selected panel of clinically relevant genes in the non-WNT/non-SHH MB subgroup are summarized in Figure 4k. Homologous recombination mutations were enriched in subtype II (*n* = 2, 50%), and DNA damage checkpoint mutations were enriched in subtype VIII (mutation in ATM: *n* = 3, 60%) (Figure 4l). C11orf95-RELA fusion was found in one non-WNT/non-SHH, and PVT1 fusion was found in subtype II (*n* = 2, 50%), III (*n* = 1, 33.3%), and IV (*n* = 1, 20%) (Appendix A). Cell type enrichment analysis found that NK and NKT cells were enriched in subtype II (Figure 4m) and were correlated with poor outcomes in non-WNT/non-SHH MBs (Appendix A). The result was validated in the SickKids cohort (Appendix A).

## 4. Discussion

In a genomic study of a large cohort, four distinct molecular subgroups of MB were identified and reported: WNT, SHH, G3, and G4 [30]. These subgroups presented specific demographics, histology, metastatic status, and prognosis [31]. By integration of gene expression and DNA methylation profiles, various subtypes with distinct demographic and clinical features were identified [2]. G3 and G4 subgroups exhibit similarities in molecular and biological profiling and are formally defined as non-WNT/non-SHH MB, which comprise eight subtypes [4]. In this cohort, 70 MBs in children and infants were retrieved and subjected to RNA-Seq and DNA methylation array analysis. SNF method and random forest scores were applied for subtype clustering to refine genetic and cytogenetic landscape within subtypes. The demographic, clinical annotations, molecular, and immune features of MB subtypes in our cohorts are summarized in Table 1.

Upon the clinical results, no metastasis presented in WNT, which was associated with very good outcomes (Figure 1g–i). Females mainly exhibited WNT in our cohort (Figure 1c and Figure 2c), however, the male/female ratio was approximately 0.8 in other studies [2,6]. Classic histology mainly existed in WNT in our cohort and the other study (Figure 1e and Figure 2e) [2]. We observed the CTNNB1 mutation in all WNT (Figure 2h), and monosomy 6 in all WNT α, as previously described (Figure 2g) [2,32,33]. DDX3X mutation existed in approximately half of WNT α, which was consistent with a previous study (Figure 2h) [32]. Interestingly, only one male WNT patient presented with recurrence.

SHH β typically occurs in children and is associated with better outcomes in our cohort (Figure 3d,h,i, and Appendix A). However, SHH β occurred in infants and was associated with worse outcomes in the other studies [2,6]. A clinical trial study reported that iSHH-II (equivalent to SHH α and SHH γ) had improved survival with reduced intensity therapy compared to iSHH-I (equivalent to SHH-β) [34]. Another clinical trial study enrolled infants with DNMB/MBEN histology, which is the majority in SHH γ, but it was closed prematurely due to an excess of relapses [35]. The treatment strategy remains a key factor to affect the prognosis in SHH MBs.

TP53 mutation in the R273H point was associated with poor follow-up in one SHH α in this cohort according to the previous study [8]. The TP53 mutation was enriched in SHH α, which was associated with worse outcomes [2,6]. The R273H mutation in TP53 can develop highly metastatic tumors in mice models [36,37]. MYCN and GLI2 were frequently co-amplified in our cohort and other cohorts [38]. MYCN and GLI2 amplification are risk factors for SHH MBs [28,39]. Consequently, treatment with SMO inhibitors by targeting MYCN and GLI2 in the SHH pathway might have efficacy for SHH MBs, which exhibited these two co-amplifications [40,41,42,43]. No PTEN deletion was identified in SHH β, whereas the deletion existed in SHH α and γ in our cohort (Figure 3d,k). SHH β presented with focal PTEN deletion, which is associated with high metastatic rates and worse survival [2].

In this study, we found that M2 macrophages and their associated genes: CCL2, CD68, CD163, CD204, CD206, CD209, CSF1R, and Dectin-1, were enriched in SHH β (Figure 3n, Appendix A, Appendix A and Appendix A). The enrichment of M2 macrophages and their associated gene expression were correlated with favorable outcomes of SHH MB in our and the SickKids cohort (Figure 3o, Appendix A, Appendix A and Appendix A). SHH MBs have significant immune signatures of T cells, fibroblasts, and macrophages [44]. During tumorigenesis, hypoxia induces angiogenesis and recruits immune cells, such as macrophages, to initiate a pro- or anti-tumor response in the tumor microenvironment (TME) [45]. The increased M2 macrophages in SHH MBs might be due to the increased CCL2, a neuroinflammatory cytokine, which could recruit and promote M2 macrophage polarization [29,46]. The infiltration of tumor-associated macrophages (TAMs) and increased expression of their associated genes, CD163 and CSF1R, were significantly observed in SHH MBs [47]. An increase in M1 macrophages was reported to correlate with good outcomes of MB [48]. Furthermore, macrophage reduction in TME were correlated with poorer outcomes of SHH MBs, and TAMs might be involved in inhibiting tumor growth in SHH MBs [46]. However, another study demonstrated that recruitment of M1 macrophages was correlated with poor outcomes of SHH MBs [49]. The above contrary studies illustrate the inconclusive and incomplete roles of TAMs to promote or suppress tumor growth in TME.

Among non-WNT/non-SHH MBs, subtype IV occurred in younger patients with a median age of 3 years and was associated with favorable outcomes in our cohort and the other study (Figure 4c,g,h, and Appendix A) [6]. Subtype VIII occurred in older children with a median age of 10 years and enriched i17q in our cohort and the other study (Figure 4c,i, and Appendix A) [6]. MYC amplification was reported as a risk factor for G3, which was mainly enriched in subtype II, which was associated with poor outcomes in our cohort and other studies (Figure 4g,h,j) [28,50]. MYCN amplifications were predominantly found in subtype V, followed by subtype VI, which was associated with poor outcomes (Figure 4g,h,j and Appendix A) [6,50]. However, MYCN amplification was not found in the only subtype V patient (Figure 4j). CDK6 amplification, which was predominantly found in G4 MBs highly enriched in subtype VI in our cohort [3,51] (Figure 4j).

In this study, we found NK and NKT cells enriched in subtype II (Figure 4m). NK cells can migrate to TME and exhibit cytolytic activity to kill tumor cells directly without specific immunization. NK cells were found to exist in MBs in the previous studies [44,52,53,54]. It was reported that NK cells can suppress SHH MB tumor growth in a Daoy xenografted mouse model [53]. NKT cells can only recognize glycolipids or lipid antigens presented by CD1d, which is a monomorphic class I HLA molecule. The expression of CD1d was reported in GBM and SHH MBs and could be a potential target for NKT cell immunotherapy [55,56]. On the other hand, some studies reported that MB can suppress NK cell attacks with TGF-β, which is an immune suppressive strategy used by tumor cells [57,58,59,60]. Therefore, subtype II MBs may secrete TGF-β to fight against the cytotoxicity of NK or NKT cells in TME and promote tumor progression. The roles of NK or NKT cells in the tumorigenesis of non-WNT/non-SHH tumors need further study.

C11orf95-RELA fusion was observed to be associated with recurrence and a poor prognosis in our cohort. C11orf95-RELA fusion, which acts as an oncogene to drive tumorigenesis through activating NF-κB signaling was identified in the majority (70%) of supratentorial ependymomas [61]. C11orf95-RELA fusion also was found to exist in ATRT [62], glioma [63,64]. C11orf95-RELA fusion-positive ependymomas associated with poor outcomes [65]. The compounds, which target NF-κB, RTK, HDAC signaling, and proteasome inhibitors could be potential drugs for C11orf95-RELA fusion-positive tumors [66].

## 5. Conclusions

In conclusion, we highlighted genomic diversities in MB subtypes in a cohort series in Taiwan. We combined two platforms: gene expression and DNA methylation profiles, for MB subtype clustering. Genetic aberrations and prognosis within subtypes were identified. We found high enrichment of M2 macrophages, and their associated genes may be an indicator of a favorable prognosis in SHH MBs. TAMs might be a therapeutic target to improve the prognosis of SHH MBs. These results will contribute to the establishment of a nationwide molecular diagnosis linked to a prognostic indicator of relevance for MBs in children.

## Figures and Tables

**Figure 1 cancers-14-05419-f001:**
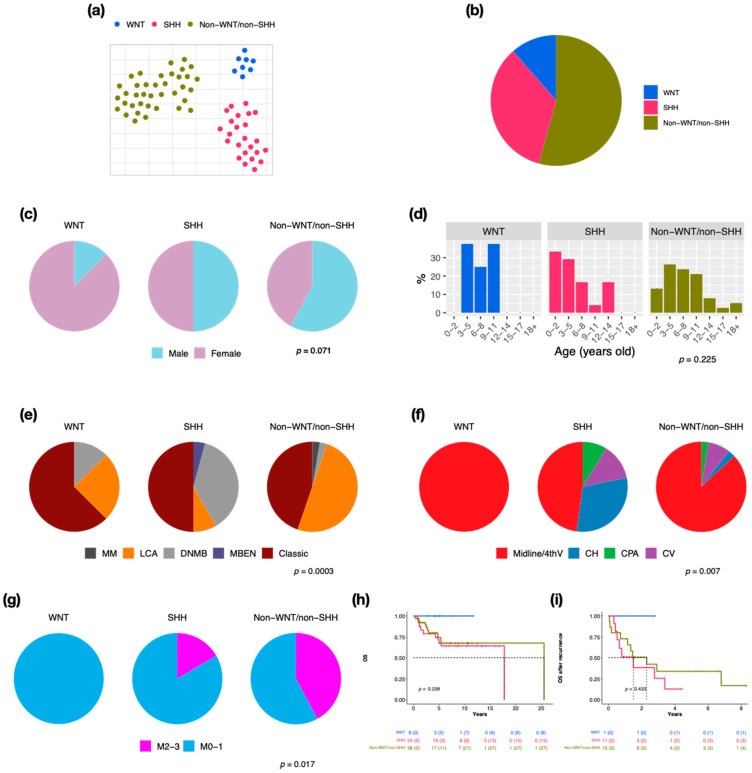
Subgroup classification, sex, and age distribution in a cohort series of 70 cases in children. t-SNE dimensional distribution (**a**) and proportion (**b**) of subgroups. Sex (**c**), age (**d**), histology (**e**), tumor location (**f**), metastasis (**g**) distribution of subgroups. DNMB: desmoplastic/nodular medulloblastoma, MBEN: medulloblastoma with extensive nodularity, LCA: large-cell/anaplastic, MM: medulloblastoma with melanotic myogenic differentiation, Midline/4thV: midline of the fourth ventricle, CV: cortex-centered including cerebellar vermis, CH: cerebellar hemisphere, CPA: cerebellar pontine angle. *p* value calculated by Fisher’s exact test. Overall survival (OS) (**h**) and overall survival after tumor recurrence (**i**) across subgroups. *p* value calculated by log-rank test.

**Figure 2 cancers-14-05419-f002:**
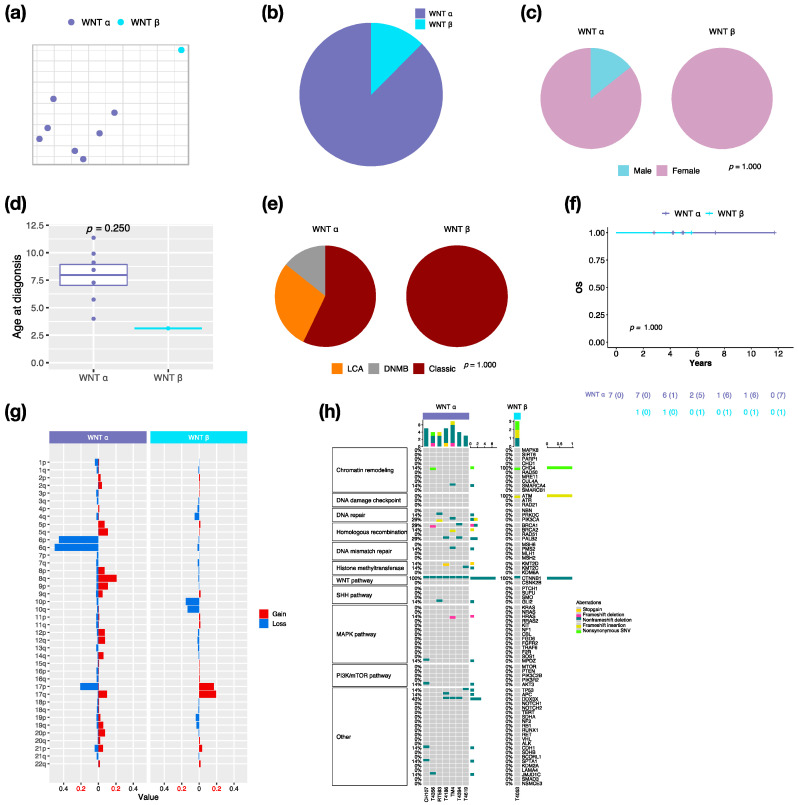
Characteristics in WNT MB subtypes. t-SNE dimensional distribution (**a**) and proportion (**b**) of WNT subtypes. Clinical features including gender (**c**), age (**d**), histology (**e**) in WNT subtypes. DNMB: desmoplastic/nodular medulloblastoma, LCA: large-cell/anaplastic. *p* value calculated by Fisher’s exact test. (**f**) overall survival (OS) across WNT subtypes. *p* value calculated by log-rank test. (**g**) Chromosomal aberrations in WNT subtypes. (**h**) Distribution of somatic mutations in WNT subtypes.

**Figure 3 cancers-14-05419-f003:**
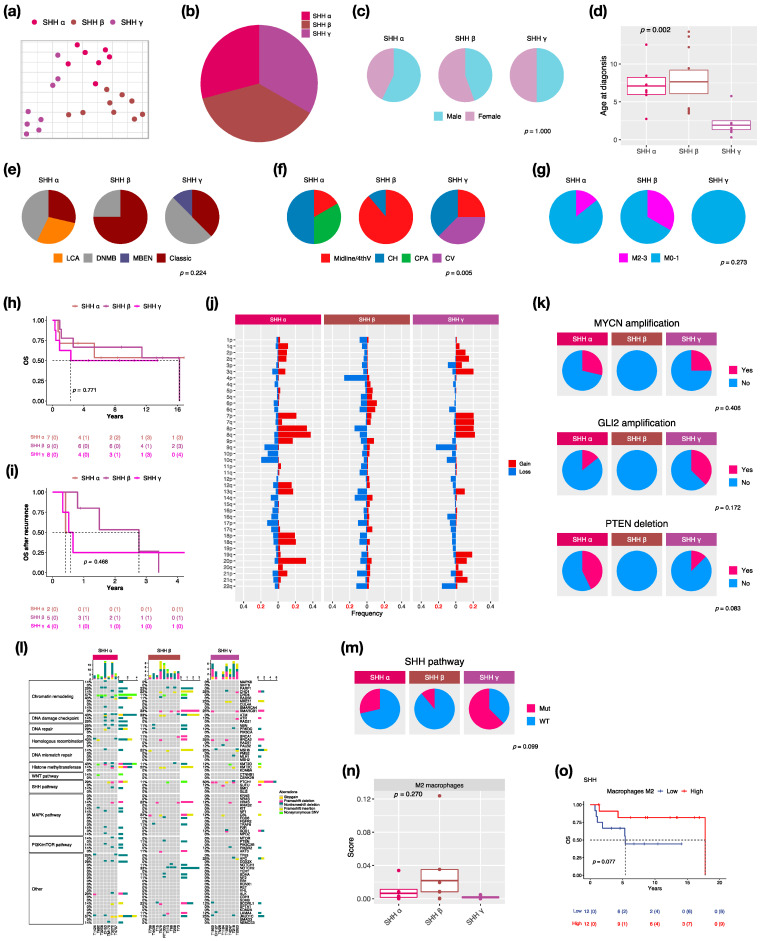
Characteristics in SHH MB subtypes. t-SNE dimensional distribution (**a**) and proportion (**b**) of SHH subtypes. Clinical features including gender (**c**), age (**d**), histology (**e**), tumor location (**f**), metastasis (**g**) in SHH subtypes. DNMB: desmoplastic/nodular medulloblastoma, MBEN: medulloblastoma with extensive nodularity, LCA: large-cell/anaplastic, Midline/4thV: midline of the fourth ventricle, CV: cortex-centered including cerebellar vermis, CH: cerebellar hemisphere, CPA: cerebellar pontine angle. *p* value calculated by Fisher’s exact test. Overall survival (OS) (**h**) and survival after tumor recurrence (**i**) across SHH subtypes. *p* value calculated by log-rank test. (**j**) Chromosomal aberrations in SHH subtypes. (**k**) Distribution of focal MYCN, GLI2, and PTEN events in SHH subtypes. *p* value calculated by Fisher’s exact test. (**l**) Distribution of somatic mutations in SHH subtypes. (**m**) Distribution of SHH pathway mutation in SHH subtypes. *p* value calculated by Fisher’s exact test. (**n**) Distribution of M2 macrophages in SHH subtypes. *p* value calculated by Kruskal–Wallis test. (**o**) OS based on high or low M2 macrophages infiltration in SHH subgroup. *p* value calculated by log-rank test.

**Figure 4 cancers-14-05419-f004:**
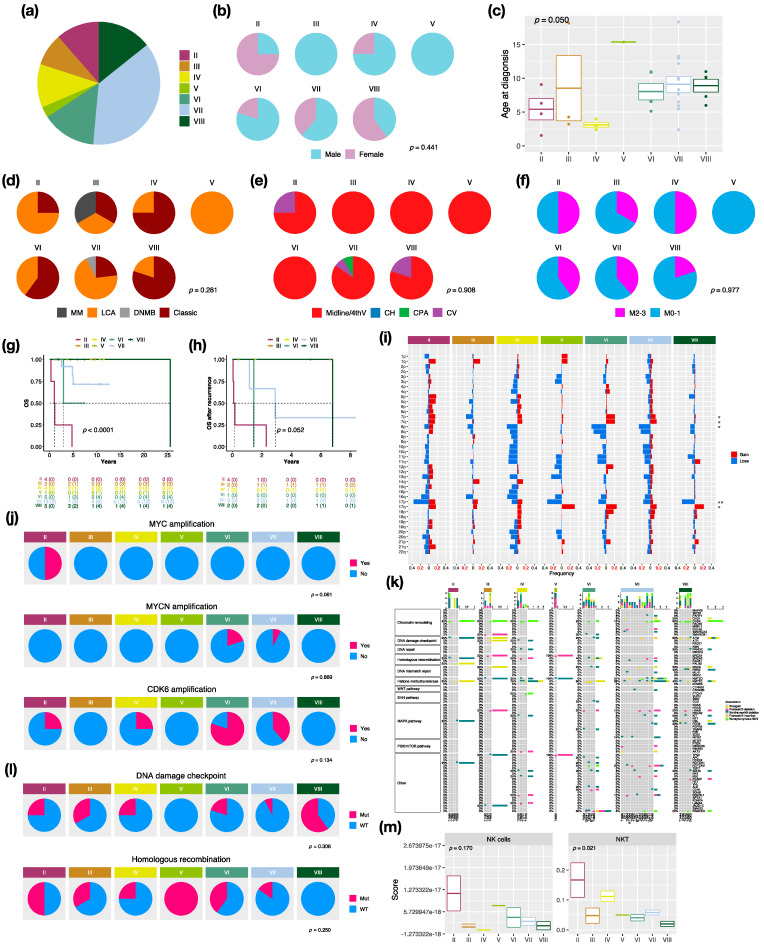
Characteristics in non-WNT/non-SHH MB subtypes. (**a**) the proportion of non-WNT/non-SHH subtypes. Clinical features including gender (**b**), age (**c**), histology (**d**), tumor location (**e**), metastasis (**f**) in non-WNT/non-SHH subtypes. DNMB: desmoplastic/nodular medulloblastoma, LCA: large-cell/anaplastic, MM: medulloblastoma with melanotic myogenic differentiation, Midline/4thV: midline of the fourth ventricle, CV: cortex-centered including cerebellar vermis, CH: cerebellar hemisphere, CPA: cerebellar pontine angle. *p* value calculated by Fisher’s exact test. Overall survival (OS) (**g**) and survival after tumor recurrence (**h**) across non-WNT/non-SHH subtypes. *p* value calculated by log-rank test. (**i**) Chromosomal aberrations in non-WNT/non-SHH subtypes. *, *p* < 0.05; **, *p* < 0.01. (**j**) Distribution of focal MYC, MYCN, and CDK6 events in non-WNT/non-SHH subtypes. *p* value calculated by Fisher’s exact test. (**k**) Distribution of somatic mutations in non-WNT/non-SHH subtypes. (**l**) Distribution of DNA damage checkpoint and homologous recombination mutations in non-WNT/non-SHH subtypes. *p* value calculated by Fisher’s exact test. (**m**) The distributions of infiltrating NK and NKT cells in non-WNT/non-SHH subtypes. *p* value calculated by Fisher’s exact test.

**Table 1 cancers-14-05419-t001:** The demographic, clinical annotations, molecular and immune features of MB subtypes in our cohorts.

Subgroup	WNT		SHH			Non-WNT/non-SHH						
Subtype	α	β	α	β	γ	II	III	IV	V	VI	VII	VIII
Frequency (%)	87.5	12.5	29.2	37.5	33.3	11.4	8.6	11.4	2.9	14.3	37.1	14.3
Median age (years)	8.4	3.1	6.5	4.1	1.4	5.5	4.3	3.0	15.4	6.6	8.3	10.1
Male/female ratio	0.2		1.3	0.8	1	0.3		3		4	1.6	0.7
Metastasis (%)	0	0	14.3	33.3	0	50.0	33.3	50.0	0	40.0	38.5	20.0
Pathology variant (%)												
Classic	57.1	100	28.6	77.8	37.5	25.0	33.3	75.0	0	60.0	23.1	80.0
DNMB	14.3	0	42.9	22.2	50.0	0	0	0	0	0	7.7	0
MBEN	0	0	0	0	12.5	0	0	0	0	0	0	0
LCA	28.6	0	28.6	0	0	75.0	33.3	25.0	100	40.0	69.2	20.0
MMMB	0	0	0	0	0	0	33.3	0	0	0	0	0
5-year OS (%)	100	100	85.7	77.8	62.5	0	100	100		50.0	71.8	100
Molecular features	CTNNB1 and DDX3X mutation, monosomy 6		MYCN amplification, PTEN deletion, TP53 mutation		MYCN, GLI2 amplification, SHH pathway mutation	MYC amplification, homologous recombination mutation				MYCN, CDK6 amplification		MYCN, GLI2 amplification, i17q
Immune features				M2 macrophages infiltration		NK, NKT infiltration						

## Data Availability

RNA-seq data are available in Gene Expression Omnibus (GSE143940 and GSE158413).

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
