# Peer review of "Clinical and Molecular Features in Medulloblastomas Subtypes in Children in a Cohort in Taiwan"

_cancers, 2022, doi:10.3390/cancers14215419_

Round 1

Reviewer 1 Report

The authors performed a molecular analysis of a cohort of taiwanese pediatric MB treated in two centers. The goal was to translate already established subgrouping/-typing to this cohort and further categorize with molecular genetic methods. As a result f.e. enrichment of SHHß MB with M2 macrophages correlated to a better prognosis was detected.

As the authors performed survival analyses, the median follow-up time of surviving patients should be added. It would also be good to give the reader a short overview of the treatment of MB at the two centers in Taiwan. Further it would be of interest to discuss, how the advanced risk risk stratification may be used to improve MB treatment in Taiwan in the future.

The current versions of the manuscript is difficult to read because there are numerous grammatical errors. It needs editing by a native English speaker.
For example the sentence: 'The results may contribute to the establishment of correlated optimal diagnosis and prognostic indicators for MBs' is not clear. I would suggest f.e.: establishment of a molecular diagnosis linked to prognostic factors of relevance for MB.
In the Discussion I would suggest to first present the results and then discuss them in the context of the current literature.

Author Response

Comment 1: As the authors performed survival analyses, the median follow-up time of surviving patients should be added. It would also be good to give the reader a short overview of the treatment of MB at the two centers in Taiwan. Further it would be of interest to discuss, how the advanced risk risk stratification may be used to improve MB treatment in Taiwan in the future.

Response: The median follow-up time of 12 subtypes were summarized in Table S1. For survival curves, the median survival was added in Figure 1h-i, 3h-i, 3o, 4g-h, S3b, S4b, S5b, S7b-c. The treatment protocol selection for MB in Taiwan was based on the adjusted Heidelberg risk stratification scheme which defined in our previous study (Cancers (Basel). 2020 Mar 11;12(3):653.) (Line 89-92, page 2).

Comment 2: The current versions of the manuscript is difficult to read because there are numerous grammatical errors. It needs editing by a native English speaker.
For example the sentence: 'The results may contribute to the establishment of correlated optimal diagnosis and prognostic indicators for MBs' is not clear. I would suggest f.e.: establishment of a molecular diagnosis linked to prognostic factors of relevance for MB.

Response: Thank you for the suggestion. 'The results may contribute to the establishment of correlated optimal diagnosis and prognostic indicators for MBs' has changed to 'The results will contribute to theestablishment of a molecular diagnosis linked to prognostic factors of relevance for MBs.' (Line 40-41, page 1; Line 53-55, page 2; Line 100-102, page 3; Line 418-420, page 14). The grammatical errors in sentences in the manuscript have been revised (Line 37-39, page 1; Line 50-52, page 1-2; Line 63-66, page 2; Line 78-85, page 2; Line 87-93, page 2; Line 97-102, page 2-3; Line 178-181, page 4; Line 207-211, page 6; Line 223-225, page 7; Line 239-241, page 8; Line 243-247, page 8; Line 249-252, page 8; Line 281-283, page 10; Line 334-335, page 12; Line 339-340, page 12; Line 342-344, page 12; Line 360-364, page 12-13; Line 386-388, page 13;).

Comment 3: In the Discussion I would suggest to first present the results and then discuss them in the context of the current literature.

Response: Thank you for the suggestion. The paragraph in Discussion were revised as results following by discussing with current literature (Line 342-344, page 12; Line 360-369, page 12-13; Line 382-385, page 13; Line 389-392, page 13; Line 401-404, page 13).

Reviewer 2 Report

A very interesting analysis of medulloblastomas (MB) submitted to RNA sequencing and DNA methylation array. During the last years, considering mainly the 2016 and 2021 WHO classifications, the diagnostic and therapeutic management of pediatric MB has been greatly modified, according to molecular characterization. The authors' manuscript contributes to a better understanding of the heterogeneity of molecular features according to subtypes' specific diagnosis and prognosis markers. 

My only issue is related to the clarity of the manuscript: to increase the readers' experience, also considering the complexity of the figures, I suggest introducing a schematic figure or table to summarise the most relevant and novel findings

Author Response

Comment: My only issue is related to the clarity of the manuscript: to increase the readers' experience, also considering the complexity of the figures, I suggest introducing a schematic figure or table to summarise the most relevant and novel findings

Response: Thank you for the suggestion. The demographic, clinical annotations, molecular and immune features of MB subtypes were summarized in Table 1 (Line 409-411, page 13-14).